# The Effect of Assistive Devices on the Accuracy of Fitbits in Healthy Individuals: A Brief Report

**DOI:** 10.3390/ijerph22071100

**Published:** 2025-07-12

**Authors:** John Jaworski, Brian Makowski, Michael Weaver, Michael Seils, Jennifer L. Scheid

**Affiliations:** 1Department of Physical Therapy, Daemen University, Amherst, NY 14226, USA; 2Department of Rehabilitation Science, University at Buffalo, Buffalo, NY 14214, USA

**Keywords:** wearable technology, Fitbits, assistive devices, walker, cane, physical activity, rehabilitation

## Abstract

Our study explored the accuracy of Fitbit recorded step count during the use of an assistive device (two-wheeled walker and standard cane) at various body positions (wrists, hips, and ankles). Participants (*n* = 11) ambulated an obstacle course (50 m total, including turns and a step up/down) a total of three times (two-wheeled walker, standard cane, and a deviceless control trial). Fitbit generated step counts (at the wrists, hips, and ankles) were then compared to the individual’s “actual” step count captured through video analysis. During the cane trial and the deviceless trial, all positions were significantly correlated (r = 0.764 to 0.984; *p* < 0.006) with the actual step count. However, increased variability (demonstrated by increased limits of agreement) was observed when the Fitbit was worn on the wrist (compared to the hips or ankles). During the walker trial, the step count was significantly correlated to the actual step count at the ankle and hip positions (r = 0.669 to 0.888; *p* < 0.017) with an average error of 1.5%, while it was not statistically correlated at the wrist with a 31.2% average error. Our study suggests that Fitbits are a good predictor of actual step count, with the caveat that the location of the Fitbit should be considered if an assistive device (e.g., two-wheeled rolling walker and single-point cane) is being used.

## 1. Introduction

The use of wearable technology has increased over the years to track activity and assess vital signs during purposeful exercise or 24 h monitoring through the objective measuring of calories burnt, steps taken, and heart rate [1]. However, the rise in general use of this technology leads to questions of accuracy, specifically in populations that utilize assistive devices during ambulation. A plethora of studies have investigated the ability of wearable technology to track steps with the healthy individual in mind [1,2,3], and a handful of smaller studies have investigated wearable technology use in older adults [4,5], patients with slower walking speeds [6,7], patients who have suffered from a stroke [8], and patients using assistive devices (e.g., walkers and canes) [9]. Gait patterns of those using assistive devices can be significantly deviated from that of an uninvolved individual, making any data collected from step tracking technology potentially unreliable [9]. With less than half (45.2%) of those living with a mobility disability reporting any participation in weekly aerobic activity, decreased means to measure such activity stand to further disadvantage a vulnerable population [10]. Compounded by the fact that tracking devices are extremely prevalent in more modern fashion and pop culture, unreliable results with such technology symbolize yet another hurdle to societal inclusivity. This will continue to be prevalent in society, as a reported 2.5 billion people are expected to use some form of assistive device [11].

The “smartwatch” industry is saturated with several devices that claim to most accurately measure one’s daily activity. While step counting may seem like a simple premise at the surface level, there are several factors that can significantly impact the accuracy of results. The placement of wearable technology at the hip, wrist, or even the ankle may impact the accuracy of steps recorded in healthy people, and reliance on an assistive device adds another variable to further complicate the equation [6,12]. It is not surprising that the accuracy of wearable technology decreases while using crutches or walkers [5,12]. While Simpson et al. demonstrated that decreased walking speeds also decreased the accuracy of wearable technology, accuracy was improved when the wearable device was worn on the ankle [6]. Kooner et al. demonstrated (with healthy participants) that the Fitbit Zip (measured at the hip) and the Fitbit Alta (measured at the wrist) had poor reliability when using assistive devices [12]. In the current study, we want to replicate this study and also examine the reliability of wearable technology at the ankle, while using a new model of Fitbit, the Fitbit Inspire 3, a device designed to measure activity while being worn at the hip or on the wrist. Additionally, we wanted to mimic daily movement patterns by creating an obstacle course, examine a two-wheeled walker (commonly used assistive device), and calculate the actual step count using video analysis.

In the current study, we systematically evaluate the accuracy of different wearable technology placements while participants use assistive devices. Specifically, the purpose of this study is to evaluate the effect of assistive devices (two-wheeled rolling walker and single-point cane) on the accuracy of Fitbits in healthy individuals on the hip, ankle, and wrist for counting step counts.

## 2. Materials and Methods

### 2.1. Participants

Eleven participants were recruited for this study via advertisements through a university listserv. Inclusion criteria for participation included being at least 18 years of age, having the ability to tolerate low to moderate physical activity, and being able to walk at least 150 m. Exclusion criteria included the following: (1) having experienced a back or lower extremity orthopedic injury (i.e., back, hip, knee, ankle, foot) that kept the individual from physical activity for at least 2 days in the past 6 months, (2) having had recent surgery to the back or a lower extremity, (3) having a diagnosed balance disorder, or (4) having a neuromuscular disorder (cerebral palsy, multiple sclerosis, spinal cord injury, traumatic brain injury, Duchenne muscular dystrophy, amyotrophic lateral sclerosis/Lou Gehrig’s disease). The participants were not currently using assistive devices. The Daemen University Institutional Review Board approved the study, and documented informed consent was obtained from all the participants prior to participation in the study.

### 2.2. Data Collection

Participants self-reported their age, gender, height, weight, hand dominance, and academic major (if applicable). Body mass index (BMI) was calculated using weight (kg)/height (m)^2^.

Each participant underwent a brief instructional session highlighting a step-through gait pattern utilizing both a single-point cane and a two-wheeled walker. During this session, concurrent and extrinsic feedback was given to ensure that all participants were able to accurately emulate the desired gait pattern prior to further data collection.

Participants were also fitted with a Fitbit Inspire 3 on each wrist, hip, and ankle. For wrist application, the supplied “watch style” band was used. Furthermore, the device came with silicon clips in which the Fitbit could be inserted into and then fastened to the waistband of a participant’s shorts or pants (for hip application). For the ankle application of the Fitbit, the silicon clips (that are typically used to fasten to a waistband) were fastened around elastic straps that were placed on each ankle.

Participants of the study then completed three trials (two-wheeled walker, standard cane, and a deviceless control trial) where they navigated a 50 m course (Figure 1). The course consisted of straightaways, turns, platform negotiation, and a five-cone weave. The three trials were completed in a random order of (1) walker–cane–no device, (2) walker–no device–cane, (3) cane–no device–walker, (4) cane–walker–no device, (5) no device–cane–walker, or (6) no device–walker–cane. Each trial was video recorded to be analyzed for actual step count.

### 2.3. Measurements

During each trial, step count was measured by six Fitbit Inspire 3 (Fitbit, San Francisco, CA, USA) devices at the six body positions (right and left wrists, right and left hips, and right and left ankles). At the end of each trial, the step count was calculated (the end step count was subtracted from the initial step count). Actual step count was visually analyzed from video recordings of each trial. One researcher (consistent throughout the study) reviewed the videos for the actual step count.

### 2.4. Data Analysis

Descriptive statistics were used to examine the participants’ demographic characteristics and were reported as mean ± standard deviation or frequency (%). The Kolmogorov–Smirnov test was used to determine if the data were normally distributed. Since some of the step counts were not normally distributed, nonparametric statistics were used. Spearman’s rank correlation measured the strength and direction of the association between Fitbit step count and actual step count, i.e., video step count. Mean observed difference (the actual step count minus the Fitbit step count) was reported, and limits of agreement (LOA) were calculated as follows: LOA = mean observed difference ± 1.96 × standard deviation of observed differences. Bland–Altman plots were generated using the mean difference and the LOAs as the reference lines. Average error (%) was calculated as [(measured step count–actual step count)/actual step count] × 100. These numbers were then averaged across body position and location. The significance level of *p* < 0.05 was used to identify all significant differences, and all data were analyzed using the SPSS (version 28.0; Chicago, IL, USA) statistical software package.

## 3. Results

### 3.1. Demographic Characteristics

Eleven participants completed the study. The participants were 24.9 ± 7.9 years old, had a BMI of 24.9 ± 4.0 kg/m^2^, 63% were male, and 73% were enrolled in a graduate physical therapy program. The mean steps of each trial were calculated as no device (77.7 ± 7.6 steps), cane (89.0 ± 6.6 steps), and walker (74.0 ± 7.8 steps).

### 3.2. Step Count with No Assistive Device

During trials with no device, all the Fitbit locations significantly correlated (r = 0.764 to 0.984; *p* < 0.006) with actual step count (Table 1). Mean observed differences (actual steps minus Fitbit steps) with no device for each position were right ankle (−1.7 ± 1.5 steps), left ankle (−2.0 ± 1.5 steps), right hip (0.2 ± 1.5 steps), left hip (0.4 ± 2.1 steps), right wrist (2.2 ± 4.0 steps), and left wrist (2.5 ± 4.8 steps). Overall LOAs for no device trials were calculated between −7.0 and 11.9 steps (Table 1, Figure 2). The average error across all participants and locations during the no device trials was 0.1%.

### 3.3. Step Count with a Single-Point Cane

During trials using a single-point cane, all positions were significantly correlated (r = 0.884 to 0.984; *p* < 0.001) with the actual step count (Table 2). The mean observed differences with a cane for each position were right ankle (−2.9 ± 2.5 steps), left ankle (−2.3 ± 2.0 steps), right hip (−0.9 ± 1.3 steps), left hip (0.3 ± 1.2 steps), right wrist (−3.7 ± 6.8 steps), and left wrist (0.1 ± 9.5 steps). LOAs while using a cane were calculated between −7.8 and 2.6 steps at the ankle or hip. However, increased variability was observed from step counts at the wrists with LOA calculated between −18.6 and 18.8 steps (Table 2, Figure 3). The average error across all participants and locations during the cane trials was −1.9%.

### 3.4. Step Count Accuracy with a Two-Wheeled Walker

The walker trial showed that the ankle and hip locations were significantly correlated (r = 0.669 to 0.888; *p* < 0.017) with actual step count. However, the step count collected at the wrist positions were not correlated (r = 0.001 to 0.206; *p* > 0.554) with the actual step count (Table 3).

The mean observed differences with a walker for each position were right ankle (−2.4 ± 4.0 steps), left ankle (−1.0 ± 3.5 steps), right hip (0.8 ± 2.5 steps), left hip (0.9 ± 2.0 steps), right wrist (82.3 ± 13.2 steps), and left wrist (88.2 ± 7.3 steps). LOAs while using a walker were calculated between −10.1 and 5.7 steps at the ankle or hip. However, the LOA was larger when at the wrist position and was calculated between 56.3 and 108.2 steps (Table 3, Figure 4). The average error across all participants and locations during the walker trials was 31.2%. However, when calculating the error during only the hip and ankle trials (when using the walker), the average error was 1.5%.

## 4. Discussion

In the current study, Fitbit Inspire 3 generated step counts were correlated with actual step counts when no assistive devices were used. More importantly, this study demonstrated that, while using a cane, Fitbit-generated step counts were also correlated at all Fitbit positions (ankle, hip, and wrist). However, increased variability (demonstrated by increased LOA) was observed when the Fitbit was worn on the wrist (compared to the hips or ankles). Lastly, this study demonstrated that, while using a walker, Fitbit-generated step count was correlated with actual step counts when the device was worn at the ankles or at the hips, but it was not correlated with actual step counts when the Fitbit was worn at the wrists. Our study suggests that Fitbits are a good predictor of actual step count, with the caveat that the location of the Fitbit should be considered if an assistive device (e.g., two-wheeled rolling walker and single-point cane) is being used.

Kooner and colleagues previously demonstrated that the Fitbit Zip (measured at the hip) and the Fitbit Alta (measured at the wrist) had poor reliability when using assistive devices; however, the measurements at the hip were more accurate than the measurements at the wrist, indicating that movement is more difficult to detect by Fitbit at the wrist when an assistive device is being employed [12]. In the current study, we also demonstrated that wearing the Fitbit at the wrist is the least accurate location to wear a Fitbit when using assistive devices. The majority of steps were not recognized by the Fitbit when worn at the wrist while using a walker, and there was increased error when the Fitbit was worn at the wrist when using a cane. In the current study, the Fitbit Inspire 3 was used, which is the updated version of both the Fitbit Zip (measured at the hip by Konner at al. [12]) and the Fitbit Alta (measured at the wrist by Konner et al. [12]). The Fitbit Inspire 3 is designed to be worn either at the hip or on the wrist with the option of placing the device into a clip (for use at the hip) or attached to watch bands (for use at the wrist). While our study did demonstrate increased accuracy (and decreased LOA) compared to Kooner et al. [12], simply using more updated technology could increase the accuracy of the step counts. Unfortunately, the reliability and validity of wearable technology is device-specific, and assessments of current devices can quickly become out of date [1]. Additionally, the current study also included the measurement of steps by attaching the Fitbit Inspire 3 to the ankle, and the ankle also provided accurate step counts while using a walker or a cane.

One of the commonly cited causes for wearable technology step count error when patients use assistive devices is the potential impact on gait speed that assistive devices impose on healthy individuals. Other studies analyzing Fitbit accuracy in healthy people using assistive devices have found an underreporting of step counts and have accredited the misrepresentation to alterations in gait pattern (caused by an unfamiliarity with such equipment in healthy people), which slow overall gait speed [12]. This theory is further bolstered by the work of Simpson et al. [6], who found that a similar model of Fitbit smartwatch reached 90% accuracy at predicting actual step count when donned at the wrist only when the wearer surpassed gait speeds of 0.8 m/s, which was the fastest time required to reach such an accuracy threshold of any Fitbit location. As such, it is possible that unfamiliarity with the assistive devices and consequential diminished gait speed throughout a participant’s ambulation in the current study could have also contributed to the error in step count measurements.

Another potential explanatory factor for the discrepancies found between Fitbit-generated step count and actual step count lies in how the assistive device impacts the gait cycle. Sala et al. [13] found a similar trend of extremely poor correlation between wrist-donned Fitbit-generated step count and actual step count in a study conducted in children with cerebral palsy and accredited the underreporting of steps to the lack of motion occurring at the wrist whilst the participants ambulated with a walker [13]. Wrist-based Fitbits may under-report step count because of the static wrist position during the gait cycle. Ambulating with one’s wrists fastened to a two-wheeled walker impedes the Fitbit’s accelerometers of sagittal plane input occurring at the wrist during typical gait, potentially explaining why, in several instances in our study, wrist-donned Fitbits failed to record steps whatsoever when used in trials utilizing a two-wheeled rolling walker.

The underreporting of physical activity (as demonstrated in the current study by underreporting step count when wearing the wearable device on the wrist while using a walker) could make using wearable technology to monitor, prescribe, or manage physical activity very difficult. Consistent underreporting could lead to frustrated patients who may actually meet their goals. Additionally, patients could overcorrect their physical activity and attempt to participate in more physical activity than is safe for them simply because their wearable technology is underreporting their steps.

Limitations of this study include both possible setup and recording errors. The startup of the Fitbits required inserting baseline biometric values such as height, weight, and sex. We entered a height of 5′9”, a weight of 160 pounds, and sex as male in order to encompass a typical person. As the Fitbits require tracking angular motion, an expected change in lever arms that are associated with higher variability due to limb lengths may skew data if one is moving their longer and shorter levers more or less when ambulating. Another limitation of this study is the subjective viewpoint of what counts as a step. Little shuffle movements when turning or stopping in the course had to be subjectively named as a step or not by our recorder. This makes data harder to record because of the step discrepancy, as Fitbits may not register these smaller movements. Another limitation is that this study was conducted with participants who were young and healthy. Using a geriatric or neurological population would be more generalizable to the populations that could benefit from the findings in this study.

Future studies should continue to investigate the accuracy of the Fitbit related to body placement (specifically hips and ankle) in other clinical populations, such as geriatric or neurological trials, as assistive devices are more prevalent in these populations. Research could further analyze how common postural deficits of geriatric patients including speed, stance phase, or swing phase gait deviations may play a factor in the reliability of assistive technology’s ability to track step count. The additional assessment of patients with neurological conditions could include individuals’ post-stroke with hemiparesis or hemiplegia, postural rigidity, shuffling, or festinating gait in individuals with Parkinson’s disease.

## 5. Conclusions

Given that the prominence of ambulatory aids in society is likely to increase as the average age of our population continues to rise, having a dependable method to measure activity in populations of people reliant on assistive devices for mobility grows increasingly more relevant. The current study demonstrated that, while using a cane, Fitbit-generated step counts were also reliable at all Fitbit positions (ankle, hip, and wrist); however, increased variability was observed when the Fitbit was worn on the wrist (compared to the hips or ankles). Additionally, the current study demonstrated that, while using a walker, Fitbit-generated step count was accurate when the Fitbit was worn at the ankle or at the hip, but it was not accurate when the Fitbit was worn at the wrist, suggesting that Fitbits are a good predictor of actual step count, with the caveat that the location of the Fitbit should be considered if an assistive device is being used. Specifically, this study suggests that the Fitbit Inspire 3 should be placed at the hip or the ankle if a walker is being used, and hip or ankle placement should be considered if a cane is being used.

## Figures and Tables

**Figure 1 ijerph-22-01100-f001:**
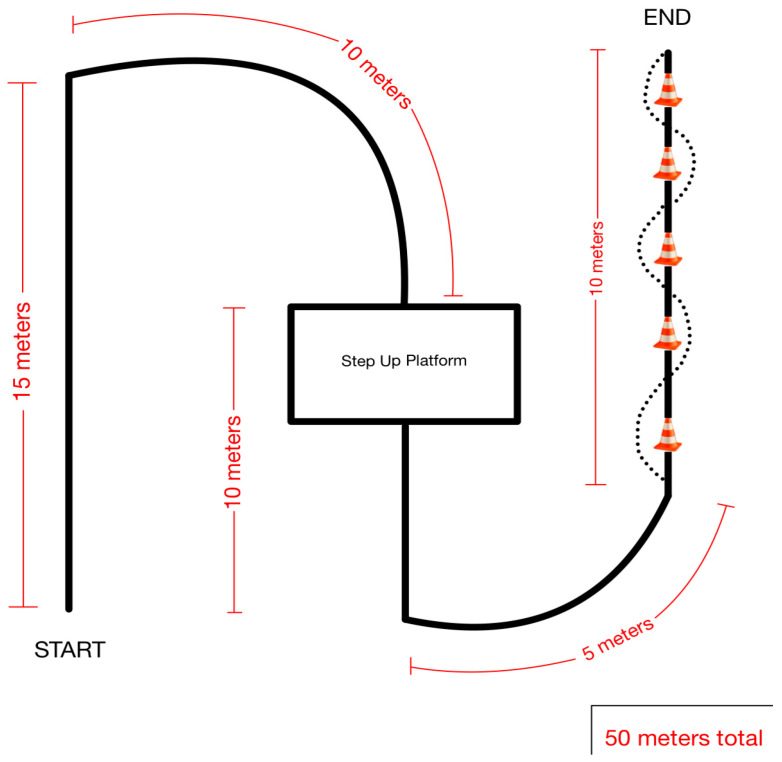
The course (50 m) ambulated by each participant three times (two-wheeled walker, standard cane, and a deviceless control trial) in a random order.

**Figure 2 ijerph-22-01100-f002:**
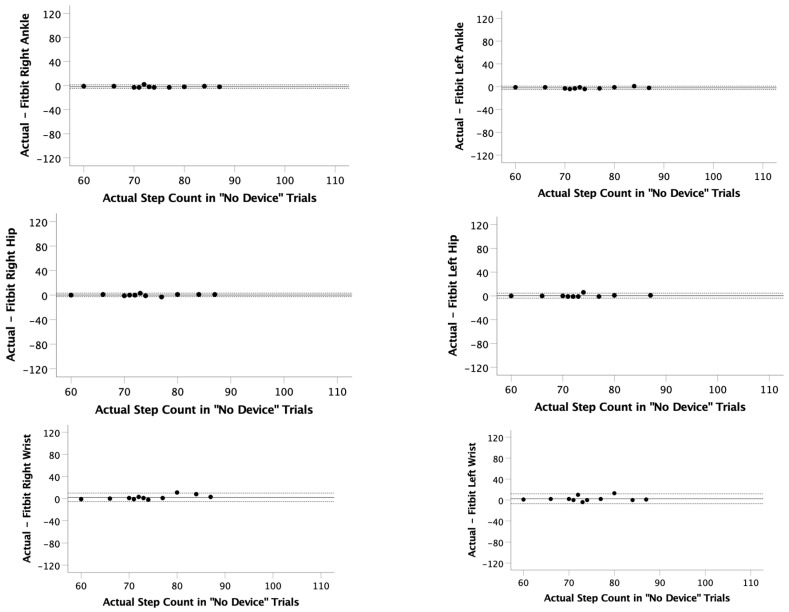
Bland–Altman plots for the “No Device” trials. The actual step count was recorded via video and compared to the step count recorded on 6 Fitbits (right ankle, left ankle, right hip, left hip, right wrist, and left wrist). Limits of agreement (LOA) were calculated as LOA = mean observed difference ± 1.96 × standard deviation of observed differences.

**Figure 3 ijerph-22-01100-f003:**
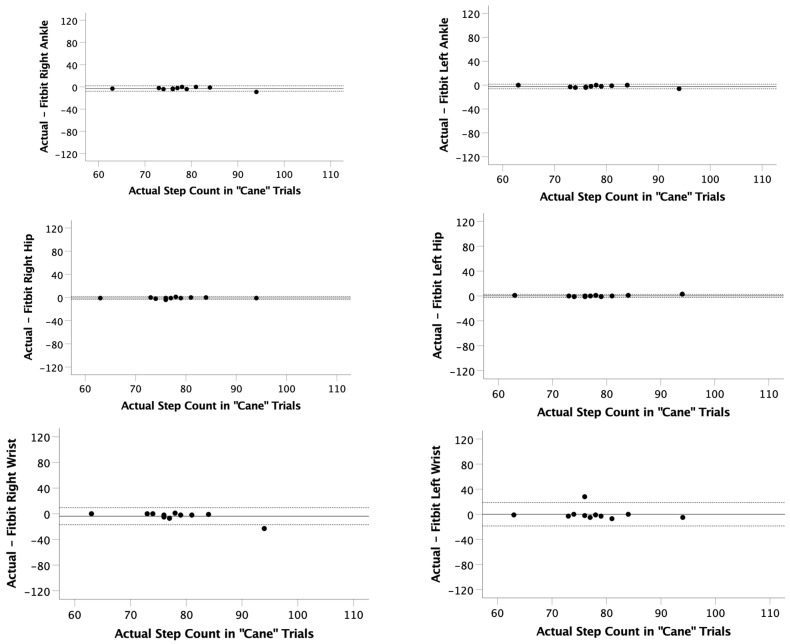
Bland–Altman plots for the “Cane” trials. The actual step count was recorded via video and compared to the step count recorded on 6 Fitbits (right ankle, left ankle, right hip, left hip, right wrist, left wrist). Limits of agreement (LOA) were calculated as LOA = mean observed difference ± 1.96 × standard deviation of observed differences.

**Figure 4 ijerph-22-01100-f004:**
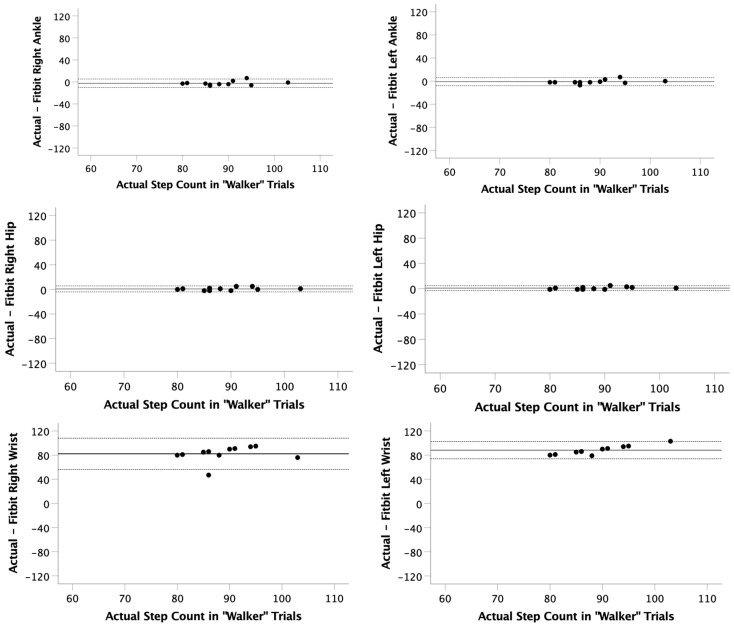
Bland–Altman plots for the “Walker” trials. The actual step count was recorded via video and compared to the step count recorded on 6 Fitbits (right ankle, left ankle, right hip, left hip, right wrist, left wrist). Limits of agreement (LOA) were calculated as LOA = mean observed difference ± 1.96 × standard deviation of observed differences.

**Table 1 ijerph-22-01100-t001:** Spearman correlations and limits of agreement when comparing Fitbit step count to actual step count in “No Device” trials.

	Actual Step Count (Video)			
	Correlation		LOA (Steps)	
	r_s_ Value	*p* Value	Low	High
Fitbit-Ankle				
Right	0.973	<0.001 *	−4.6	1.2
Left	0.970	<0.001 *	−5.0	1.0
Fitbit-Hip				
Right	0.934	<0.001 *	−2.8	3.2
Left	0.897	<0.001 *	−3.8	4.6
Fitbit-Wrist				
Right	0.817	0.002 *	−5.6	10.0
Left	0.764	0.006 *	−7.0	11.9

* *p* < 0.05 considered significant. “No Device” trials: No assistive devices were used, and the actual step count was recorded via video and compared to the step count recorded on 6 Fitbits (right ankle, left ankle, right hip, left hip, right wrist, and left wrist). Limits of agreement (LOA) were calculated as LOA = mean observed difference ± 1.96 × standard deviation of observed differences.

**Table 2 ijerph-22-01100-t002:** Spearman correlations and limits of agreement when comparing Fitbit step count to actual step count in “Cane” trials.

	Actual Step Count (Video)			
	Correlation		LOA (Steps)	
	r_s_ Value	*p* Value	Low	High
Fitbit-Ankle				
Right	0.899	<0.001 *	−7.8	2.0
Left	0.908	<0.001 *	−6.1	1.6
Fitbit-Hip				
Right	0.929	<0.001 *	−3.5	1.6
Left	0.984	<0.001 *	−2.1	2.6
Fitbit-Wrist				
Right	0.884	<0.001 *	−17.1	9.6
Left	0.895	<0.001 *	−18.6	18.8

* *p* < 0.05 considered significant. “Cane” trial: A cane was used, and the actual step count was recorded via video and compared to the step count recorded on 6 Fitbits (right ankle, left ankle, right hip, left hip, right wrist, and left wrist).

**Table 3 ijerph-22-01100-t003:** Spearman correlations and limits of agreement when comparing Fitbit step count to actual step count in “Walker” trials.

	Actual Step Count (Video)			
	Correlation		LOA (Steps)	
	r_s_ Value	*p* Value	Low	High
Fitbit-Ankle				
Right	0.699	<0.017 *	−10.1	5.4
Left	0.725	<0.012 *	−8.0	6.0
Fitbit-Hip				
Right	0.883	<0.001 *	−4.0	5.7
Left	0.888	<0.001 *	−3.0	4.8
Fitbit-Wrist				
Right	0.206	0.554	56.3	108.2
Left	<0.001	1.00	73.9	102.5

* *p* < 0.05 considered significant. “Walker” trial: A walker was used, and the actual step count was recorded via video and compared to the step count recorded on 6 Fitbits (right ankle, left ankle, right hip, left hip, right wrist, and left wrist).

## Data Availability

The data presented in this study are available upon request from the corresponding author.

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
