# Peer review of "The Effect of Assistive Devices on the Accuracy of Fitbits in Healthy Individuals: A Brief Report"

_ijerph, 2025, doi:10.3390/ijerph22071100_

Round 1
Reviewer 1 Report
Comments and Suggestions for Authors
The paper is well-written and clearly presented. The analyses are appropriate. That being said, the study was very similar to that of Kooner et al (ORR 2021, Volume 13, 1–8, 312) – both studies used small samples of healthy adults. The major differences were that a newer model of Fitbit was evaluated, a measurement course which more likely mimics daily movement patterns was used, dominant/non-dominant sides were not evaluated, fewer types of assistive devices were used, and an additional attachment site was used (ankle). Thus, this study does address some aspects not present in the Kooner et al. study. That being said, the major issue with the Kooner et al. paper (which the authors mention as a limitation) was that healthy individuals comprised the sample. This major limitation is repeated in this paper (healthy, young). I would be of the view that to significantly advance the knowledge in this field, at least an elderly healthy sample could have been included. I would suggest this paper be reformatted to a Brief Report.
Reviewer 2 Report
Comments and Suggestions for Authors
The authors aimed to examine the accuracy of the Fitbit Inspire 3 when measuring step counts at various locations, with different walking aids when compared to visually recorded steps. The authors concluded that the Fitbit Inspire 3 is accurate when worn at the ankle or waist, but not at the wrist. The study was generally conducted well, and the manuscript is well written. Below are some general and specific comments I hope will help to improve the manuscript further.
Introduction
- I would suggest the rationale for the study should be more explicit. Research has already been done on the accuracy of a range of step count devices with the use of different walking aids. Is the aim to replicate/further add to this data, or specifically to investigate the locations that have not been investigated i.e. the ankle?
Methods
- Page 3, line 107 – Was actual step count analysed by one or multiple investigators? Ideally two investigators would both visually count steps and agree on a number for each trial. Please may you make it clearer if this was done, or if it was done by a sole researcher.
- I would suggest constructing Bland-Altman plots with Limits of Agreement to display the results visually, rather than just having the limits of agreement in the tables, as it will make interpreting easier. You could also add in any bias the Fitbit displays. I would also suggest adding a table or graph with mean step counts from each location during each trial.
- How was error calculated, as it is not described in the data analysis section, only that mean observed difference was calculated. Was it Mean Percentage Error or Mean Absolute Percentage Error? For transparency, please add the formula used.
Results
- Page 5, line 173 – How did you determine the agreement was ‘poor’? Are there predefined values for limits of agreement that would class agreement as excellent/good/moderate for example?
Discussion
- I would suggest placing greater emphasis on the implications of the findings. The underreporting of wrist worn devices is inherent to walking aids, as gait and speed will always be different from unaltered walking. Given the importance of physical activity for health, consistent underreporting of step counts will make prescribing/managing activity difficult, and to ensure those with walking aids are fully benefitting, accurate equipment is needed. Perhaps a few sentences to this effect would be helpful here and would link to your recommendations in the conclusion.
- Page 7 – Were participants already using walking aids or not? Based on the methods, where you state participants were instructed on a correct gait, I would also suggest a limitation is that the participants were not people who require walking aids. This may not reflect the natural gait of habitual walking aid users.
Great work, and good luck!
Reviewer 3 Report
Comments and Suggestions for Authors
This manuscript presents a focused, well-structured experimental study investigating how assistive device use (cane, two-wheeled walker) and Fitbit placement (wrist, hip, ankle) affect the accuracy of step counting in healthy adults.
- Scientific Merit and Originality
- Weaknesses:
- The sample is limited to 11 healthy, young-to-middle-aged individuals (many of whom are PT students), limiting generalizability to elderly or clinical populations.
Future studies should consider larger, more diverse populations (e.g., elderly, neurologically impaired), different device brands, and gait speed as a covariate.
- Methodology and Data Analysis
- Weaknesses:
- Use of uniform biometric input (height, weight, sex) across all devices could introduce bias due to different limb lengths or body mechanics.
Include a justification or sensitivity analysis to discuss the potential impact of entering identical demographic data into all devices.
- Results Interpretation
- Results clearly show that:
- Hip and ankle placements provide accurate step counts in all conditions.
- Wrist placement is unreliable, especially during walker use.
- The 32.9% error during walker use at the wrist is important and well-highlighted.
Clarify in the discussion that wrist-based Fitbits may over-report due to static wrist positions on walkers (as seen with inflated values).
- Literature Review and Referencing
- Gaps Identified:
- There is no mention of machine learning approaches used in modern gait analysis or device-specific calibration techniques which could be highly relevant.
- The role of gait speed thresholds (e.g., <0.8 m/s) in device reliability could be more deeply explored using wearable AI/ML models.
Suggested References to Add:
Del Din, S., et al. (2016). “Free-living monitoring of Parkinson’s disease: lessons from the field.” Movement Disorders.
Also mention the other wearable devices such as IMUS and provide pros and cons against fitbits etc. or advise on joint use for wider applications etc.
Simultaneous validation of wearable motion capture system for lower body applications: over single plane range of motion (ROM) and gait activities, BIOMEDICAL ENGINEERING-BIOMEDIZINISCHE TECHNIK, 2022, 0013-5585, 67, 3, 185-199.
Also you may make use of the Bland Altman Graphs and provide an explanation over systematic bias and random variations and explaining number of points falling out of upper and lower boundary limits as in the below and above references
Wearable Motion Capture System Evaluation for Biomechanical Studies for Hip Joints’ Journal of Biomechanical Engineering, APRIL 2021, Vol. 143 / 044504-12021
Also mention for the adaptability of the device for being applicable to other daily life activities for counting by referring to the below paper
'Database covering the prayer movements which were not available previously ' NATURE SCIENTIFIC DATA, 2023, 2052-4463, 10, 1. v
- Clarity and Structure
- There is a slight inconsistency in labeling (e.g., "trail" instead of "trial" in some places).
- Table legends can be improved by emphasizing statistical relevance in plain
Proofread for minor typos and label consistency (e.g., “trail” → “trial”), and clarify figure legends for broader accessibility.
Comments on the Quality of English Language
revision is required
Round 2
Reviewer 1 Report
Comments and Suggestions for Authors
The authors have sufficiently addressed my concerns. As a Brief Report, this paper would be of interest to researchers wishing to extend the knowledge in this area.